# FAST POST-TRAINING ANALYSIS OF NERFS USING A SIMPLE VISIBILITY PREDICTION NETWORK

## ABSTRACT

Exercising NeRFs on real-world data taught us that their novel view rendering capability varies across different views and rendering of regions that are visible in more input images often produces more reliable results. However, efficient quantitative tools haven't been developed in this regard to facilitate the post-training analysis of NeRF rendered images. In this paper, we introduce a simple visibility prediction network that efficiently predicts the visibility of *any* point in space from *any* of the input cameras. We further introduce a visibility scoring function that characterizes the reliability of the rendered points, which assists the evaluation of NeRF rendering quality in the absence of ground truth. Utilizing this tool, we also empirically demonstrate two downstream post-training analysis tasks. The first task is to reduce rendering artifacts via modified volumetric rendering which skips unreliable near-range points. We achieve an average PSNR improvement of 0.6 dB in novel view rendering without changing the network parameters of the pre-trained base NeRF on a benchmark composed of 62 scenes. The second task is to select additional training images to re-train a NeRF and enhance its rendering quality. By re-training the base NeRF with a handful of additional views selected using the proposed visibility score, we achieve better rendering quality compared to random selection. Our method is rudimentary, yet efficient and simple to implement making it a suitable drop-in tool for various post-training tasks beyond the studies shown in this paper.

## 1 INTRODUCTION

Training NeRFs involves minimizing the empirical photometric errors between pixels in input images and predicted ones. Consequently, each pixel is "explained" by multiple 3D points along the ray passing through the pixel, and each 3D point must explain its pixel observation in one or more input views. The emergence of scene "geometry" in NeRFs therefore can be considered as the process of discovering non-transparent points in the space that can consistently explain observations across one or multiple views. As the outcome of NeRF based scene reconstruction, one may imagine that view synthesis of highly visible areas is expected to produce more reliable results. This phenomenon is well understood from a machine learning perspective: the mean squared photometric error calculated by projecting points along a ray to one pixel is statistically a biased estimator of the expected photometric error per 3D point rendered at a novel view. The less number of input views a point is visible from, the stronger such bias is. In an extreme case, optimizing a point to explain as few as only one input view leads to lower training errors than optimizing a point to explain multiple. Yet, such a point which fits better in training often generalizes poorly to novel views.

Motivated by the fact that such novel view generalization capability correlates with how many input views a point is visible from, we advocate to perform post-training analysis of NeRFs to examine the visibility of all rendered points from a given novel view. In particular, we develop an efficient tool to quantify the risk of rendering a low-visible point in novel view synthesis. Calculating the transmittance of a point from all training cameras is a computationally demanding process as one has to calculate the footprints of the same point from all training views using volumetric rendering. To alleviate the heavy computation involved in estimating point visibility, our core idea is to develop a visibility prediction network that is trained concurrently with the NeRFs at small overheads. Given $K$ training cameras, the visibility prediction network (VPN) takes a point in $\mathbb{R}^3$ as input and outputs a $K$ dimensional vector, each element of which corresponds to the logit of probability of the point

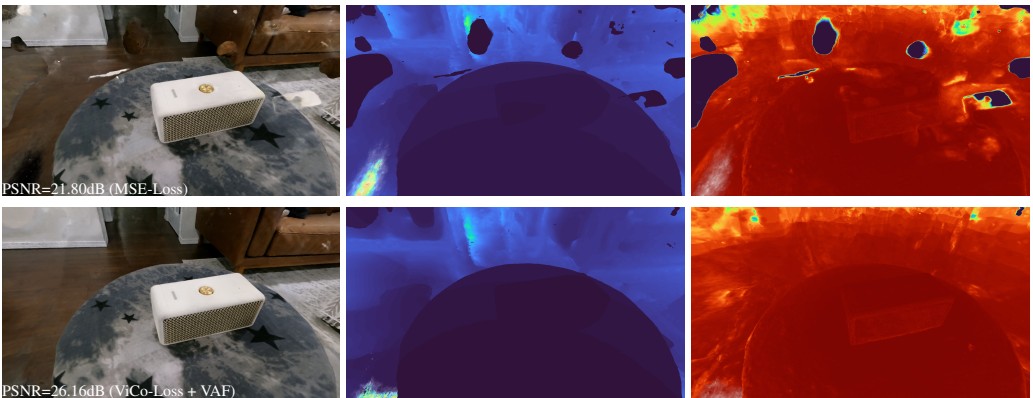

Figure 1: Columns from left to right: image rendered at a novel view, predicted depth map, visibility score $\tau(n_{\mathbf{P}}^{\text{pred}})$ (cold color indicates low visibility from training views). Rows from top to bottom: Nerfacto, Nerfacto rendered with visibility analysis and filtering (Sec. 3.1). Note, all models are trained with the same set of 50 input views only to amplify the known limitations of NeRFs.

being visible from the respective input camera. Furthermore, by aggregating visibility information from all input views for each point in space, we derived a point based visibility score – an uncertainty based measure that quantifies the reliability of a point rendered by a pre-trained NeRF. The first row in Figure 1 is a visualization of our visibility measure calculated based on VPN. As one may observe those high-risk areas (cold color in third column) correlate well with (i) artifacts or "floaters" widely known in the NeRF literature, and (ii) regions that were not sufficiently observed in input images (e.g. far-range background).

To showcase the utility of our proposed visibility prediction network and the image uncertainty derived from the point visibility scoring, we demonstrate two post-training applications for NeRFs.

- The first application is to identify and remove artifacts in NeRFs that are trained on object centric views. We developed a simple heuristic that filters any low-visible near-range point in volumetric rendering. We show that this simple heuristic is effective in removing significant amount of floaters in NeRFs and thus improves the PSNR metric by a large margin regardless how NeRFs were trained in the first place. In particular, we demonstrate the effectiveness of such floater removal on a large scale NeRF benchmark composed of 62 object scans acquired in 6 different indoor and outdoor environments.

- The second application is to aggregate visibility scores over all pixels in any novel view of interest, and evaluate how useful it can be to enhance the performance by re-training NeRFs with additional views. Capturing images for NeRF production in high standard filming and entertainment is known to be a tedious task: a one-time capturing effort of data is often not adequate for producing high fidelity rendering at a user specified trajectory. Allowing one to re-train a NeRF on multiple rounds of data acquisition could significantly benefit the novel view synthesis performance if those additional views collected compensate the limitation of the previous collections. The visibility toolbox we developed in this paper is therefore highly useful in such multi-session data acquisition process. As a proof of concept, we demonstrate through an experiment that by carefully adding a handful of extra views to the original training set, one can significantly boost the performance of a pre-trained NeRF through re-training on the augmented training set.

To summarize our contribution, (i) this is the first work to systematically perform visibility analysis of a reconstructed scene based on implicit neural representations. (ii) We employ an efficient network to model the visibility field and predict the visibility score of any 3D point in the scene observed from any of the training views. We envision that real-world applications of NeRFs, for instance, NeRFs for visual effects, will require one to characterize the quality of the rendering in the absence of ground-truth data, and the proposed visibility prediction network serves this purpose. (iii) Furthermore, our work allows for a number of downstream post-training applications that aim

to enhance the quality of novel view rendering. We demonstrated two of such applications through empirical evaluation.

The rest of paper is organized as follows: Sec. 2 discusses the expanded and highly evolving literature around NeRF and specifically focuses on differentiating our work from other related work that adopt similar techniques. Sec. 3 describes the proposed visibility prediction network and its implementation details. Sec. 4 describes two experiments that demonstrated the usefulness of our approach. Sec. 5 concludes our paper by discussing potential future work and a few preliminary results that we believe are worth sharing with the community.

## 2    RELATED WORK

**NeRF and its improvements.** Since the original NeRF Mildenhall et al. (2021) has been invented, a number of methods have been developed to further improve its quality in novel view rendering. For example, Mip-NeRF Barron et al. (2021) proposed to, for each pixel, sample points along a viewing frustum instead of a single ray passing through the pixel to address the aliasing problem. A continued work, Mip-NeRF360 Barron et al. (2022) further addressed some technical limitations of Mip-NeRF, for instance, contracting and modeling unbounded scenes.

One issue Mip-NeRF360 along with several other work called out is the presence of the undesirable blurry "floaters" formed as part of NeRF training. Mip-NeRF360 proposed a distortion loss that regulates the density distributions of particles along each sampled ray to partially alleviate this problem. Some other work observed that the floaters often appear in regions close to the camera (near-range regions), and as such referred the floaters as "background collapses". They proposed methods to avoid the formation of floaters by either scaling down the gradients Philip & Deschaintre (2023) or enforcing sparsity of the density field Yang et al. (2023) in near-range regions. There are other regularization methods that leverage learned priors. For example, Roessle et al. (2022) proposed to use depth priors estimated from images to assist the optimization of radiance fields. In Warburg et al. (2023), 3D diffusion priors are introduced to regularize the density field in NeRFs. Another explanation Martin-Brualla et al. (2021); Tancik et al. (2023) for the cause of those near-range floaters is that forming the floaters right in front of the camera helps reduce the training loss when the global lighting varies in the training data – violating the constant illumination assumption made by NeRF. Incorporating an appearance embedding for each training frame helps model the inconsistent global lighting and thus remove some floaters at the cost of lower PSNR metric.

We want to mention that all the existing explanations of floater formation are consistent with our findings in this paper. The floaters, regardless of the cause of their formation, coincide with the least visible area from training views, and therefore are susceptible to overfitting leading to a higher chance to be seen and perceived as annoying floaters in the test views.

**Efficient training of implicit volumetric fields.** One limitation of the original NeRF approach is that its training often takes a huge amount of GPU-hours even for a small scale scene. Particularly, it uses a large Multilayer Perceptron (MLP) to calculate the density and view-dependent color of points. A number of methods have been developed to significantly accelerate the training. Instant-NGP Müller et al. (2022) uses multi-resolution hash grids with small MLP decoders to output density and color of points. This technique makes querying a bounded volumetric field several orders of magnitude faster and affects a sparse set of model parameters for each point. Nerfacto Tancik et al. (2023) and Zip-NeRF Barron et al. (2023) further integrates the multi-resolution hash grids along with other best practices in the literature to allow training unbounded scenes in less than an hour. In our work, we use Nerfacto as our base NeRF model and develop visibility prediction network trained concurrently with the base model. Particularly, we use a separate network instance with multi-resolution hash grid for visibility prediction (See Sec. 3).

**Post-training evaluation of NeRFs.** Evaluation of NeRFs has been dominated by ground-truth based methods which directly compare synthesized images against the acquired real images at the same camera pose. Although these ground-truth based metrics (e.g. PSNR, SSIM Wang et al. (2004), LPIPS Zhang et al. (2018)) serve sufficiently well if the purpose is to evaluate the interpolation capability of NeRFs, the real-world NeRF practices often go beyond just interpolation. One is often interested in using a NeRF to render images along a camera path that is drastically different from the camera path used in data acquisition (e.g. capture data at the ground level, and render from

a bird-eye view), or using a simulated camera model that is different from the one used in data acquisition. In those cases, one is left with little options of evaluation methodologies. In fact, we argue that visibility based post-training analysis seems to be the most straightforward choice in evaluation of a trained NeRF.

**Visibility prediction.** Using visibility prediction network to facilitate the calculation of point visibility along a ray without sampling has been explored in a few recent works. For example, Somraj & Soundararajan (2023) uses visibility prior estimated from stereo images to regulate the training of NeRFs with sparse input images. The prediction network they developed only applies to points seen by stereo images, not any point in a novel view. Tancik et al. (2022) uses visibility prediction to decide how to merge multiple NeRFs in rendering an aggregated novel view. The visibility prediction they proposed is used to differentiate different NeRF models instead of different input cameras as ours does. Srinivasan et al. (2021) also uses visibility prediction to estimate surface light visibility across various viewing angles for the purpose of relighting a NeRF-like model with global illumination. Although there are a few similar techniques that predict visibility of points, our method calculates visibility of *any point in space from* any input view which achieves the greatest flexibility and is crucial for fine-grained post-training analysis.

## 3   VISIBILITY PREDICTION NETWORK

**NeRF and visibility scores of points.** NeRFs Mildenhall et al. (2021) and its variants render a pixel based on sampled points $\{\mathbf{p}_i \in \mathbb{R}^3\}_{i=1}^n$ along a viewing ray in direction $\varphi \in \mathbb{S}^2$ using volumetric rendering: Given a density field $\sigma(\cdot) : \mathbb{R}^3 \to \mathbb{R}^+$ and a color field $c(\cdot, \cdot) : \mathbb{R}^3 \times \mathbb{S}^2 \to [0, 1]^3$, let $\sigma_i = \sigma(\mathbf{p}_i)$ and $c_i = c(\mathbf{p}_i, \varphi)$, we have

$$
\begin{aligned}
c_{\text{pred}}(\{\mathbf{p}_i\}_{i=1}^n, \varphi) &:= \sum_{i=1}^n w_i c_i + (1 - W) c_b, \\
w_i &:= (1 - \exp(-\sigma_i \delta_i)) \cdot T_i, \\
T_i &:= \exp\left(-\sum_{j=1}^{i-1} \sigma_j \delta_j\right),
\end{aligned}
\tag{1}
$$

where $c_b$ is the background color, $T_i$ is the transmittance up until $\mathbf{p}_{i-1}$, $\delta_i > 0$ is the length of the interval that includes point $\mathbf{p}_i$, and $W := \sum_{i=1}^n w_i \le 1$. The visibility of point $\mathbf{p}_i$ along this ray is therefore approximated as:

$$
v(\mathbf{p}_i) := T_i \cdot \exp\left(-\frac{\sigma_i \delta_i}{2}\right) \in [0, 1], i = 1, \dots, n.
\tag{2}
$$

One can also derive a continuous version of visibility function that is defined for any points along a ray $(\mathbf{o}, \mathbf{d})$: let $\mathbf{p}(t) = \mathbf{o} + t\mathbf{d}$ a variable point of $t \in [0, \infty)$,

$$
v(\mathbf{p}(t)) := \exp\left(-\int_0^t \sigma(\mathbf{p}(s)) \mathrm{d}s\right).
$$

Let's denote this visibility function as a field per input view defined over the frustum of camera's Field of View (FoV) $F^{(k)} \subset \mathbb{R}^3$ as $v^{(k)}(\mathbf{p})$, where $k$ represents the $k$-th training view. Particularly $v^{(k)}(\mathbf{p})$ is measured based on the ray originated from the center of the $k$-th camera and passing through point $\mathbf{p}$. Furthermore, let $v^{(k)}(\mathbf{p}) = 0$ for any out-of-FoV point $\mathbf{p} \in \mathbb{R}^3 \backslash F^{(k)}$. For the brievity of notation, given $K$ training views in total, we denote

$$
\mathbf{v}(\mathbf{p}) := \left[v^{(1)}(\mathbf{p}), v^{(2)}(\mathbf{p}), \dots, v^{(K)}(\mathbf{p})\right] \in [0, 1]^K.
$$

For a point $\mathbf{p}$ that is visible in at least one training view, we define the normalized attribution probability $\tilde{\mathbf{v}}(\mathbf{p}) = \dfrac{\mathbf{v}(\mathbf{p})}{\mathbf{v}(\mathbf{p})^T \cdot \mathbf{1}}$ (therefore $\tilde{\mathbf{v}}(\mathbf{p})^T \cdot \mathbf{1} = 1$[1]) and its effective sample size (ESS) reads

---

[1] We normalize the probability vector since we are interested in calculating the conditional probability such that $\tilde{\mathbf{v}}(\mathbf{p})_i = \text{Prob}(\mathbf{p} \text{ is visible from camera } i | \mathbf{p} \text{ is visible })$.

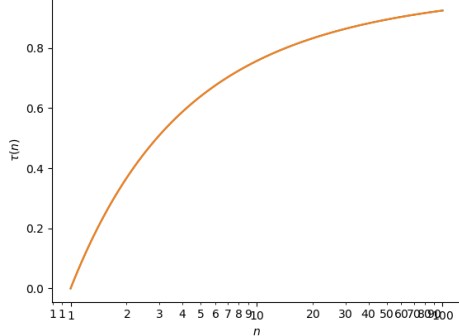

Throughout this paper, we define

$$\tau(n) := \frac{2}{n-1} \cdot \left[ \frac{\Gamma(n/2)}{\Gamma((n-1)/2)} \right]^2, \quad \forall n \in (1, \infty) \tag{4}$$

where $\Gamma(\cdot)$ is the gamma function. This formula is the bias correction multiplier applied to estimate standard deviations of normal random variables Gurland & Tripathi (1971). One can show that $\tau(1) = 0$ and $\tau(n) \to 1$ as $n \to \infty$.

Figure 2: LogX-Y plots of $\tau(n)$ in Equation 5, where $n$ is considered as the effective number of training views that sees a 3D point.

as

$$n_{\mathbf{p}} := \frac{1}{\tilde{\mathbf{v}}(\mathbf{p})^2 \cdot \mathbf{1}} = \frac{\left( \sum_{i=1}^{K} v^{(i)}(\mathbf{p}) \right)^2}{\sum_{i=1}^{K} v^{(i)}(\mathbf{p})^2}, \tag{3}$$

Remark that $\tilde{\mathbf{v}}(\mathbf{p})$ is a discrete distribution that represents how much a non-transparent point at location $\mathbf{p}$ attributes its responsibilities to explain across $K$ input views: A point that is not visible from one view has no responsibility to explain the observation from that view; a non-transparent point that is visible from all views must explain all of them. $n_{\mathbf{p}}$ is essentially the effective number of views a non-transparent point at location $\mathbf{p}$ must explain.

For sampled points $\{\mathbf{p}_i\}_{i=1}^n$ along a ray $r$, we can now conveniently define the visibility score of the ray by:

$$\tau_r := \sum_{i=1}^{n} w_i \tau(n_{\mathbf{p}_i}) \tag{5}$$

where $\tau(n)$ is defined by Equation 4 (See Fig. 2 for more information). The first row of Figure 1 depicts the relevance between $\tau_r$ and artifacts rendered in novel views by a popular NeRF method. From the visualization, we observe that near-range floaters or artifacts rendered by the NeRF coincide with the low visible region in the analysis.

Evaluating each $\tau(n_{\mathbf{p}_i})$ involves computing $\mathbf{v}(\mathbf{p}_i)$ along rays passing through the same 3D point across all $K$ training views. Since $K$ is often a few hundreds, evaluating $\tau$ thus requires several orders of magnitude more compute than standard volumetric rendering. To alleviate this issue, a natural choice is to develop efficient approximation methods for predicting $\mathbf{v}(\mathbf{p}_i)$ as is done in this paper. The approximation method reuses the pixel sampling pattern in vanilla NeRF training and trains a new network for predicting point visibility.

### 3.1 Approximation of $\mathbf{v}(\mathbf{p})$ Using Visibility Prediction

Suppose there are $K$ training views in total, and training a NeRF involves sampling rays from each of the training views. We create a separate backbone network that takes a point $\mathbf{p} \in \mathbb{R}^3$ as input and outputs a $K$ dimensional vector $\mathbf{r}_{\text{pred}}(\mathbf{p}) \in \mathbb{R}^K$, each $\mathbf{r}_{\text{pred}}^{(k)}(\mathbf{p})$ approximates the logit of $v^{(k)}(\mathbf{p})$ if $\mathbf{p}$ is within the FoV of the $k$-th camera. Given ray indexed batch samples in training NeRF, we concurrently optimize this other separate network via binary cross entropy loss: for points $\{\mathbf{p}_i\}$ sampled along a ray from the $k$-th input view, we attempt to minimize

$$\sum_{i=1}^{n} \text{BinaryCrossEntropy} \left( \left[ v^{(k)}(\mathbf{p}_i) \right]_{\text{stop\_grad}} ; \text{sigmoid} \left( \mathbf{r}_{\text{pred}}^{(k)}(\mathbf{p}_i) \right) \right), \tag{6}$$

where $v^{(k)}(\mathbf{p}_i)$ is calculated as part of volumetric rendering by Equation 2 each time a ray from the $k$-th input view is sampled, $[\cdot]_{\text{stop\_grad}}$ implies the back-propagated gradients from the cross entropy loss are stopped to not affect NeRF parameters.

For training the visibility prediction network, because we can only sample points within camera FoV, the output $\mathbf{r}^{(k)}_{\text{pred}}(\mathbf{p})$ is only meaningful if $\mathbf{p}$ falls within the FoV of the $k$-th training camera. To ensure we have reliable visibility prediction for all cameras and all points, we need another dense grid predictor $\mathbf{g}(\text{contract}(\mathbf{p})) \in [0, 1]^K$ for FoV checks. Here $\text{contract}(\cdot)$ is the contraction function that maps $\mathbb{R}^3$ or a bounded region to the normalized volume $[0, 1]^3$. This dense grid predictor $\mathbf{g}$ is computed beforehand and kept frozen in training. It uses tri-linear interpolation to predict whether a given point falls within the FoV of one training camera. We found using a coarse grid with resolution $64^3 \times K$ or $128^3 \times K$ is sufficient for our purpose. Finally, we have a composite visibility prediction function:

$$\mathbf{v}_{\text{pred}}(\mathbf{p}) := \text{sigmoid}(\mathbf{r}_{\text{pred}}(\mathbf{p})) \odot \mathbf{1}(\mathbf{g}(\text{contract}(\mathbf{p})) > 0.5) \in [0, 1]^K, \tag{7}$$

where $\odot$ denotes the elementwise product between two vectors. By replacing $\tau(n_{\mathbf{p}_i})$ in Equation 5 with $\tau(n^{\text{pred}}_{\mathbf{p}_i})$, it reads

$$\tau^{\text{pred}}_r := \sum_{i=1}^{n} w_i \tau(n^{\text{pred}}_{\mathbf{p}_i}), \quad n^{\text{pred}}_{\mathbf{p}_i} := \frac{\left(\sum_{k=1}^{K} \mathbf{v}^{(k)}_{\text{pred}}(\mathbf{p}_i)\right)^2}{\sum_{k=1}^{K} \mathbf{v}^{(k)}_{\text{pred}}(\mathbf{p}_i)^2}. \tag{8}$$

In short, the NeRF network and visibility prediction network are optimized using different loss functions and influence each other during the concurrent training. The visibility prediction network identifies the occluded regions as the predicted geometry of NeRF evolves.

## 4 APPLICATIONS OF POST-TRAINING ANALYSIS

To illustrate the usefulness of our proposed post-training visibility based analysis, we focus on describing two potential applications that haven't been done in the literature. We want to call out that these experiments we conducted are deemed proof-of-concept as they are configured in a way to illustrate the effectiveness of our method by amplifying issues observed in real world practice. Since our experiment/method is highly reproducible, we expect the same improvements can be observed in other configurations, models or datasets, yet at varied scales.

### 4.1 SKIPPING LOW-VISIBLE NEAR-RANGE POINTS IN VOLUMETRIC RENDERING

In the volumetric rendering of NeRFs, the trained model predicts a density field output $\{\sigma_i\}$ for a set of points $\{\mathbf{p}_i\}$ along the ray. For sampling based rendering methods, evaluating the density field is done in multiple rounds, often from coarser density approximator to finer ones. For ray marching based rendering methods, evaluating the density field is progressively done one point after another until the accumulated opacity is above a threshold. Our method can be applied to both families of rendering methods by immediately resetting the output $\sigma_i$ to zero if certain criterion are met. The criterion we use throughout our experiment is to check if $\tau(n^{\text{pred}}_{\mathbf{p}}) < 0.9$ and $\text{depth}(\mathbf{p}) < 1$ [2].

We use Nerfacto Tancik et al. (2023) as our base model given that it provides good trade-off between quality and training efficiency. Nerfacto uses two rounds of proposal sampling, each round of which requires to evaluate a proposal density field that outputs the coarser density estimation for a set of points. We apply our filtering criterion on both the proposal samplings and final NeRF evaluation (See Equation 1), which effectively skips any low-visible near-range points in volumetric rendering. As shown in the bottom row of Fig. 1, applying this simple filtering strategy in volumetric rendering leads us to produce less artifacts or floaters without changing any parameters of a pre-trained NeRF.

To quantitatively evaluate the performance improvement of novel view rendering, we evaluate the approach on a large benchmark called *ObjectScans* which focuses on evaluating object centric

---

[2] Assume that the entire scene is re-scaled by the largest distance between any two input cameras, as is the common practice of NeRFs.

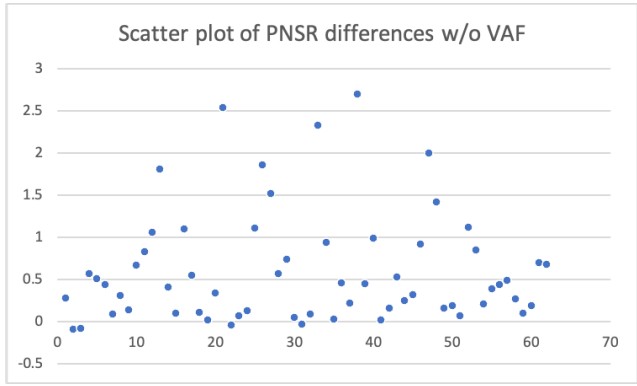

Figure 3: Scatter plot of PSNR differences w/o VAF across 62 datasets. X-axis is the dataset index, Y-axis the difference between Nerfacto w/o VAFs. The change per dataset in PSNR is in range of $(-0.091, 2.547)$. 58 out of 62 datasets observe improvements, and four observe visually indistinguishable degradation.

NeRFs. We collected this dataset using GoPro HERO 9 cameras' video mode, taken from 6 different environments. The benchmark is composed of 62 datasets (the "High Quality" subset), each capturing a common household object in an unbounded environment from circular views. Comparing most of existing public benchmark which focuses STOA NeRFs on ideal data captures, our benchmark is larger and has a wide coverage of real-world challenges in data acquisition such as motion blurs, limited viewpoints, varied lighting conditions, as well as difficulties in modeling reflective materials, low texture surfaces, and high topological complexity.

In our experimental setting, we evenly sample 50 images per dataset as the training set, and use the rest 250 images per dataset as the evaluation set. We report the averaged statistics in Table 1 and plot individual PSNR changes in Figure 3. We observe our visibility analysis and filtering (VAF) method produces rendering quality improvements across 58 out of 62 datasets in *ObjectScans* High Quality subset, with about 0.6 PSNR averaged improvements. Noticeably 12 datasets have greater than 1 PSNR improvement, while 4 datasets have small, visually indistinguishable degradation in PSNR.

| Method | PSNR ↑ | SSIM ↑ | LPIPS ↓ |
|---|---|---|---|
| Base model | 25.43 | 0.8259 | 0.2655 |
| Base model + VAF | 26.04 | 0.8327 | 0.2545 |

Table 1: Averaged metrics across 62 datasets in ObjectScans. VAF denotes visibility analysis and filtering method we proposed in Sec. 4.1.

## 4.2 SELECTING ADDITIONAL VIEWS FOR ENHANCING NERF PERFORMANCE

The visibility based post-training analysis proposed in this paper improves rendering quality by removing floaters as demonstrated in the earlier experiment, however it does not correct catastrophic failures in NeRF training, which often happens in casually captured datasets. It is well known in NeRF community that it takes significant effort to prepare a NeRF ready dataset for high quality virtual production and an amateur practitioner often needs multiple rounds of data acquisition to obtain a desirable result (i.e., clean and sharp rendering along a specified trajectory). Therefore, a much desirable envisioned application of our visibility based analysis is to provide feedback to the data acquisition process. Those feedback helps practitioners reduce their effort by focusing on capturing only minimal number of selected views with the hope that these additional data can provide information preventing catastrophic failures previously observed in trained NeRFs.

Formally speaking, given a set of candidate views $\Omega^c$ of interest sampled around a scene, one is thus interested to select the top-K useful views from $\Omega^c$ that can augment the performance of the NeRF previously trained upon a fixed set of training images $\Omega$. One may consider a few simple heuristics:

Rule 1. Skip any view in $\Omega^c$ if it's too close to a view in $\Omega$, since this new view adds little diversity in the existing pose selection.

Rule 2. If one view in $\Omega^c$ is selected, any other view that is too close to the selected view can be skipped so as to maximize the diversity in pose selection.

Practicing only rule 1 and 2 still leaves one with ample options of selecting top-K views. We demonstrate that it is helpful to use visibility analysis to determine edibility of those additional views. Particularly, for candidate set $\Omega^c$, we calculate and rank views based on the following index: for each view $I \in \Omega^c$,

$$C_I := \sum_{r \in I} \mathbf{1}(\tau_r < 0.9 \text{ and depth}_r < 1) \cdot (\text{depth}_r)^\gamma \qquad (9)$$

where $r$ denotes each ray in rendered view $I$, $(\text{depth}_r)^\gamma$ denotes the footprint weight of rendered points, and when $\gamma = 2$, the footprint weight is equal to the area. We use $\gamma = 1$ in our experiments.

Similar to our previous experiment, we use 50 evenly sampled images as the base training set $\Omega$, but use another disjoint 50 evenly sampled images as the evaluation set $\Omega^t$. The rest 200 images are considered as candidate set $\Omega^c$. We algorithmically select 10 images from 200 candidates and add them to the 50 previously selected training images, re-train the NeRF model from scratch based on the newly aggregated training set, finally evaluate the performance improvements on the evaluation set.

In the view selection process, we first filter out candidate views based on Rule 1., and calculate our proposed index $C_I$ for each $I \in \Omega^c$. We greedily select the highest $C_I$ in the remaining candidates one after another by iteratively filtering out views based on Rule 2. The index $C_I$ attempts to detect views that capture large footprint low-visible geometries in the reconstructed scene. We found that those views when added back to the training set can substantially reduce the risk of forming floaters or generating catastrophic failures in NeRF training. Figure 4 depicts some selected views from our approach as well as their visibility scoring maps. Table 2 summarizes our quantitative results by practicing the proposed view selection on 6 datasets in *ObjectScans*. As the table shows, by adding only 10 carefully selected views to training set, one can significantly improve NeRF compared to randomized selection.

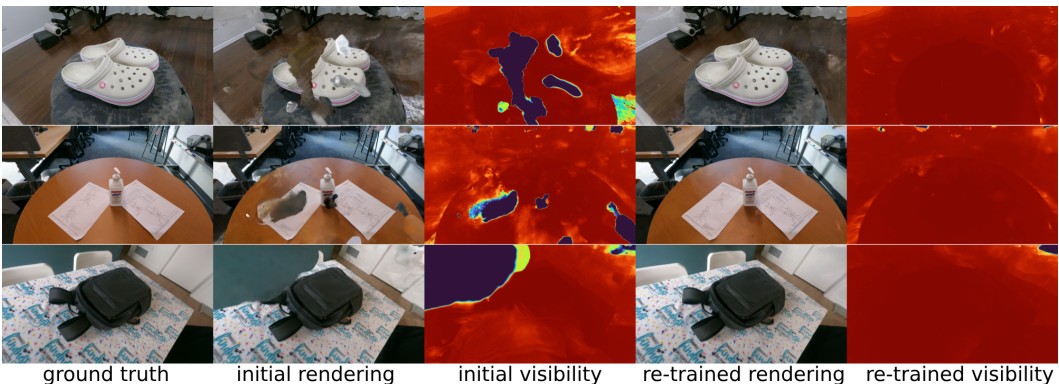

Figure 4: First column: the ground-truth images; 2nd and 3rd columns: sample of the selected views rendered based on the NeRF previously trained on 50 evenly sampled images; 4th and 5th columns: the selected view rendered based on re-trained NeRF from the additional 10 images.

## 5 FUTURE WORK AND CONCLUSION

**Future work.** We emphasize that our proposed visibility based analysis can be helpful for applications beyond the experiments demonstrated in the paper. One interesting direction is to explore in-painting for NeRF rendered images. Since a NeRF model cannot faithfully generate natural looking images in regions where it does not have sufficient input data, the rendering of those regions

| Datasets | Base model | Randomized Selection | Selection based on $C_I$ |
|---|---|---|---|
| Crocs | 25.78 / 0.883 / 0.248 | 26.12 / 0.891 / 0.242 | **27.33 / 0.904 / 0.218** |
| GoPro | 27.64 / 0.863 / **0.197** | **27.92 / 0.866** / 0.199 | 27.88 / **0.866 / 0.197** |
| InstantPot | 23.20 / 0.834 / 0.231 | 23.36 / 0.836 / 0.229 | **24.04 / 0.842 / 0.220** |
| InfantToy | 24.07 / 0.860 / 0.295 | 25.13 / 0.869 / 0.280 | **25.33 / 0.874 / 0.273** |
| Sanitizer | 26.84 / 0.890 / 0.217 | 27.22 / 0.892 / 0.213 | **28.00 / 0.900 / 0.205** |
| Schoolbag | 24.45 / 0.852 / 0.217 | 25.40 / 0.864 / 0.200 | **26.19 / 0.871 / 0.186** |

Table 2: PSNR(↑) / SSIM(↑) / LPIPS(↓) metrics on evaluation set. We empirically compare the effectiveness of the selection strategies, one based on the proposed $C_I$, one is randomized.

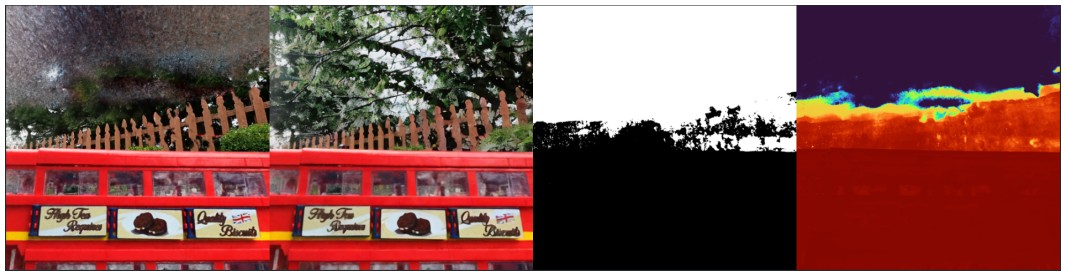

Figure 5: An example where visibility based analysis produces a confidence mask that one can inpaint low-visible background pixels to replace the synthesized pixels from NeRF. From left to right: NeRF rendered images, inpainted images using the mask; mask images derived from visibility scores; visibility scoring map. Rendered images are cropped to $512 \times 512$ to fit the input size of SD 2.0 inpainting.

often produce unfavorable results in the form of foggy clouds. Figure 5 shows a preliminary example about using Stable Diffusion 2.0 inpainting model [3] to hallucinate low visible area of a NeRF rendered image (e.g. far background) determined by our approach. Another interesting direction is to explore how to accommodate visibility analysis in regulating the training of common NeRFs, especially for sparse input view configurations or reconstruction of challenging non-lambertian materials.

**Conclusion.** Visibility, albeit probably the most obvious attribute one wants to examine in any multi-view 3D reconstruction, is very under explored for implicit neural representations. This paper lays the ground work for supporting visibility based post-training analysis for NeRFs. The method proposed is elementary and easy to implement. Besides using visibility score as an evaluation metric in the absence of ground-truth data, the paper further quantitatively demonstrates its effectiveness in two downstream applications.

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
