# OpenReview forum: "Fast Post-training Analysis of NeRFs Using A Simple Visibility Prediction Network"
_ICLR.cc/2024/Conference — Submitted to ICLR 2024_

### Official Review · Reviewer_Ekv4 · 2023-10-28

**Soundness:** 2 fair
**Presentation:** 2 fair
**Contribution:** 2 fair
**Rating:** 3
**Confidence:** 5

**Summary:**

In this paper, the authors facilitate the post-training analysis of NeRFs by presenting a visibility prediction network (VPN) that estimates the visibility of any point in space from any input camera and a visibility scoring function that evaluates the reliability of rendered points. The VPN takes a 3D point as input and outputs a K-dimensional vector, where each element represents the logit of the probability that the point is visible from the corresponding input camera.  The authors demonstrate the effectiveness of their approach on two post-training analysis tasks: 1) reducing rendering artifacts, and 2) selecting additional training images to retrain a NeRF and improve its rendering quality.

**Strengths:**

1. The paper is mostly well-written.
2. Experimental results show the effectiveness of the proposed approach on two downstream tasks.

**Weaknesses:**

The following weaknesses are ordered by importance:

1. Experimental results are insufficient. The method is only evaluated on a dataset collected by the authors. More common benchmarks should be considered for a thorough evaluation. Additionally, the authors need to better visualize and describe the collected dataset. The paper lacks free-viewpoint-rendered videos for qualitative presentation and does not compare the proposed method to other approaches for improving NeRF rendering quality. The proposed procedure is only applied to nerfacto, but not other NeRF-based methods, which limits its generalizability.

2. The proposed approach is not very convincing. The idea is straightforward and lacks novelty, and the authors do not provide a strong motivation for using the visibility score as a criterion. Equation 2 requires further explanation or derivation to enhance the reader's understanding.

3. I am not convinced of the rationale behind the task of selecting additional views.

3. The VPN needs to be trained concurrently with NeRF, which is not as simple as just a post-training process. The authors should emphasize this aspect.

4. Other minor issues: a. Most of the citations should be parenthesized. b. The numbers on the n-axis of Figure 2 overlap with each other, making it difficult to read. c. Figure 3 does not clarify the unit for PSNR and is not presented in a reader-friendly manner. d. The introduction section lacks important citations, which should be added to provide a more comprehensive background and context.

**Questions:**

1. In the abstract, it is stated that the visibility prediction network can predict the visibility of any point in space from any of the input cameras. How is this achieved? Can you explain the approach in more detail?

2. The paper mentions that "the NeRF network and visibility prediction network are optimized using different loss functions and influence each other during the concurrent training." How exactly does the training behavior of NeRF get influenced by the visibility prediction network? Can you explain this in more detail?

3. The paper proposes a method for skipping low-visible near-range points to reduce rendering artifacts. How are the visibility and depth thresholds for skipping determined? Since the visibility of points can vary significantly across different scenes, how does the proposed method handle this variability?

---

### Official Review · Reviewer_2Uqd · 2023-10-30

**Soundness:** 3 good
**Presentation:** 3 good
**Contribution:** 2 fair
**Rating:** 5
**Confidence:** 3

**Summary:**

This paper introduces a point-wise visibility prediction network (VPN) to facilitate the post-training analysis of NeRF rendering images. A visibility scoring function is also provided to characterize the reliability of the rendered points. Two downstream post-training analysis tasks that skip unreliable near-range points and select extra training images to re-train a better NeRF are illustrated for demonstration. The numerical experiment and visualizations show the efficiency of the proposed method on a self-collected dataset.

**Strengths:**

•	The paper is well motivated to the topic of visibility-based post-training analysis.

**Weaknesses:**

The experiments need to be strengthened to support the proposed method.

  * The visibility of sampled points along a given ray is defined as (2). While using (2) for guiding the binary cross entropy loss in (6), it is curious how much the extra training time compared with the original NeRF training.

  * It would be better if Figure 3 comprised two other metrics (SSIM, LPIPS).

* It would be better if the experiments were also conducted on some well-known NeRF datasets.

* In Table 1, what is the upper bound using the visibility defined in (2) rather than the predicted one?

* In Task 1, what is the performance gain compared with distortion loss?

* In Task 2, what is the performance if using Rule 1, Rule 2, randomization?

**Questions:**

The primary concern of this paper is that its experiments need to be strengthened to support the idea. Please see [Weaknesses] for more details.

---

### Official Review · Reviewer_Fmeq · 2023-11-01

**Soundness:** 3 good
**Presentation:** 4 excellent
**Contribution:** 3 good
**Rating:** 3
**Confidence:** 3

**Summary:**

This work introduces a new way to judge the confidence of a 3D point in a learnt radiance field. They propose a k-dimensional term called visibility for a 3D point which tells how clearly a point is visible from the k different training views. They provide a statistical formula for this term and show that it requires a lot of computation. As a workaround, they suggest to use a neural network to predict this term by calculating the original visibility values and regressing against them. For training purposes, the visibility network is accompanied by a dense grid which helps to determine whether a point lies within the field-of-view of the k-th camera or not. The paper showcases the benefits of their analysis using two different use-cases. The first use-case is the skipping of low-visible near-range points during rendering. The second is the selection of additional views that can improve the NeRF rendering quality. The visibility map produced by the method can also be used to remove artefacts from images. The authors show a cool application by inpainting a 2D rendered image with stable diffusion.

**Strengths:**

- The paper is well written and easy to read.
- The authors are up-to-date with the recent advances in the field and have cited them.
- The paper has targetted an important problem of judging the confidence of NeRFs from views where camera poses are not present (where metrics like PSNR, SSIM, etc. cannot be used due to lack of ground truth).
- The paper incorporates the proposed method to improve the quality of NeRF and present it using results on the Nerfacto variant. The approach presented should be applicable to almost all ray-casting based NeRF variants.

**Weaknesses:**

- The work relies on a very heavy grid alongside the neural network to predict the visibility. The paper states that they deal with datasets which have hundreds of views leading to an overwhelmingly huge grid size. I am concerned with the feasibility of their proposed method as a post-processing step given that it involves such heavy lifting.
- It seems that the authors are not releasing the dataset. Please read the clarification requested below.

**Questions:**

- The authors say that they evaluate their approach on a large benchmark called ObjectScan. The authors also say that they scan the dataset themselves. Are the authors calling their own captured dataset ObjectScan? If no, then original ObjectScan dataset is not cited. If yes, then why is the dataset not listed as a contribution? Are they not planning to release it? The authors are requested to clarify the source of the dataset and it's release conditions.
- The authors use an FoV grid which is coupled with the visibility prediction network. They state "For training the visibility prediction network, ...". Does this imply that the grid is only used for training? Is it not used for inference at all? If it is not used for inference, then how can the network predict the visibility correctly since the multiplicative term will be missing?
- If the authors are using the grids for both training and inference, then is a neural network requierd to predict the visibility? The 3D grid stores a k-dimensional vector at each location and the neural network is also predicting a k-dimensional vector and both of them are multiplied. In this scenario, the grid itself should be sufficient to learn the visibility especially since the network is taking only the position as input (and no view-direction). Grids are an explicit representation of it. Did the authors try this? (I am mentioning this approach since it seems more efficient. I have worked with this kind of setup. Hence, this question.) These aspects need to be mentioned very clearly.
- The authors should give details about the training procedure and timings for their methods. Post-processing methods are usually meant to be lightweight and fast. It doesn't seem that it is the case for this work.


Additionally, the authors should check out highly relevant concurrent works: https://bayesrays.github.io and https://repo-sam.inria.fr/fungraph/active_camera_placement. They seem to be  relevant to their work and will help in better understanding.

**Details Of Ethics Concerns:**

Nothing special in this work

---

### Official Review · Reviewer_jS1P · 2023-11-01

**Soundness:** 3 good
**Presentation:** 2 fair
**Contribution:** 2 fair
**Rating:** 5
**Confidence:** 5

**Summary:**

This paper proposes to learn an additional network to predict the visibility of each point at each input view. The predicted visibility information can be used to down-weight the 3D points with low visibility probability during the volumetric rendering or as a criterion to select additional views to retrain the nerf. The PSNR/SSIM/LPIPS are improved in these two cases.

**Strengths:**

1.  A method to distill the visibility information from density values to a lightweight network.
2.  The experimental results verify the improvement of rendering quality by integrating the visibility information.

**Weaknesses:**

1. There already exists research papers that investigate the visibility information into neural implicit representation, such as "Neural Rays for Occlusion-aware Image-based Rendering" in CVPR 2022, but not cited or discussed in this paper.


2.   A minor issue：the 3D points in the air before the object surfaces are transparent, not occluded. It might be confusing to classify them as low visibility points.

**Questions:**

Eq.6 means the visibility prediction network tries to learn how to map a 3D point to its average v(p) for all views. It resembles the behavior of integrating the density values along all rays for each 3D point. While the experimental results show that it is beneficial to the rendering quality, but it might lead to holes in the rendering results. Did you notice that in your experimental results？

---

### Meta-Review · Area_Chair_mpcW · 2023-12-04

**Metareview:**

This paper investigates how to analyze and improve the quality of NeRF rendered images. It presents two tools to achive the goal. First, it proposes a visibility prediction network to predict the visibility of points from any views, and emploies the network to select additional views to retrain NeRF. Second, it proposes a visibility scoring function to estimate the reliability of rendered points, and emploies the function to reduce rendering artifacts by removing unreliable near-range points.

Strengths:
+ It introduces a lightweight network for extracting visibility information from density values.
+ The paper is well-written.

Weaknesses:
- The novelty is limited. There is already papers that investigate the visibility information into neural implicit representation, such as "Neural Rays for Occlusion-aware Image-based Rendering" in CVPR 2022, but not cited or discussed in this paper.
- The proposed method requires huge grid size and might not feasible as a post-processing step.
- There are several related work missing.
- Experimental results are insufficient. The method is only evaluated on a dataset collected by the authors. More common benchmarks should be considered for a thorough evaluation.
- The dataset is not released for reproducing the results.
- There is not enough details to validate the proposed work. There are some details missing such as how the visibility and depth thresholds for skipping determined.

**Justification For Why Not Higher Score:**

Although the proposed idea is interesting and well-motivated,
all reviewers were in agreement that this paper is not ready for publication due to limited novelty and missing details. Further, there was no rebuttal, and the reviewers' opinions did not change after the rebuttal period. I recommend not to accept this paper at this stage, giving the authors more time to improve their paper.

**Justification For Why Not Lower Score:**

N/A

---

### Decision · Program_Chairs · 2024-01-16

Reject